# Robust Virtual Sensing of the Vehicle Sideslip Angle through the Cross-Combination of Multiple Filters Using a Decision Tree Algorithm

**DOI:** 10.3390/s23135877

**Published:** 2023-06-25

**Authors:** Gaël P. Atheupe, Younesse El Mrhasli, Ulrich Emabou, Bruno Monsuez, Kenneth Bordignon, Adriana Tapus

**Affiliations:** 1Renault Group Technical Centre, 1 Avenue. du Golf, 78280 Guyancourt, France; 2Autonomous Systems and Robotics Lab/U2IS, ENSTA Paris, Institut Polytechnique de Paris, 91120 Palaiseau, France; 3Department Aerospace Engineering, Embry-Riddle Aeronautical University, Prescott, AZ 86301, USA

**Keywords:** vehicle dynamics, sideslip angle, cross-combination, state estimation, machine learning

## Abstract

This paper presents a state-of-the-art estimation technique by cross-combining a number n of filters for high-precision, reliable and robust vehicle sideslip angle state estimation, over a full range of vehicle operations irrespective of the driving mission and disruptions that may occur in the system. A machine-learning algorithm based on decision trees connects several filters together to switch between them according to the driving context, ensuring the best possible state estimate for relatively small and large sideslip angle values. In conjunction with the above-mentioned aspects, a seamless transition between different vehicle models is attained by observing the key parameters characterizing the lateral motion of the vehicle. The tests conducted using a prototype vehicle on a snow-covered track confirm the effectiveness and reliability of the proposed approach.

## 1. Introduction

Once again, the automotive industry is gearing up towards another revolutionary move. The over-actuation of future vehicles and the smart by-wire transformation offer unique capabilities to designing new vehicle stability functions for safer and more comfortable driving such as torque vectoring (TV) and sideslip control logic (SCL) which have the potential to provide significant safety benefits in global car crashes, and nicely increase the vehicle’s steering performance. Unlike the electronic stability control (ESC), which is curative in nature, TV prevents critical driving situations by continuously monitoring and controlling the SSA. However, this parameter upon which the development of TV software relies is not readily available due to the lack of on-board measurement devices or sideslip angle (SSA) sensors in current vehicles. As a matter of fact, the interest in a real-time application to know the SSA has motivated the completion of several studies in the field of vehicle lateral motion control and SSA estimation due to the desire for a real-time application that can accurately determine the SSA [1]. Estimating the sideslip angle of a vehicle becomes particularly difficult in low grip conditions, when one or both axles slip excessively. Under normal driving conditions, a Kalman filter would generally do the job [2,3] providing a good design and the tuning of the covariances. However, to ensure the robustness and reliability of our approach, we will conduct thorough tests and validation specifically in low grip conditions. The major SSA estimation approaches presented in the literature include: the model-based method [4], the vision-based method [5], sensor fusion [6], the data driven method [7], and the hybrid method [8].

The model-based design is about the art of compromising between the level of complexity of the process model and the expected accuracy of the state estimate.

To achieve the best possible state estimate, refs. [9,10] use two different vehicle models depending on the driving context. They include the kinematic model and the dynamic model. The results show that by combining the benefits of both models, while rejecting their penalties, a decent SSA estimate can be achieved. Likewise, the authors of [11] cross-combined a KF fed by a kinematic model and a UKF using a dynamic model to ensure a good SSA estimation in the face of critical driving conditions. The emphatic point to note here is that the kinematic model with a Kalman filter (KF) is triggered through weighting terms when the vehicle speed and yaw rate values are far from zero, without which the states of the process model become unobservable. The dynamic filter that only comes into play when the yaw rate is close to zero to support the kinematic filter using a kinematic model is now unobservable. In contrast to other methods that utilize two vehicle models or two filters, ref. [12] introduced a KF-based SSA estimator that can be used across a broad range of vehicle operations. This approach achieves adaptability by adjusting the covariance matrix terms, enabling a smooth transition between different tire models. Depending on the driving situation, the linear tire model is used for reliable estimation in case of small SSA and linear tire behavior, while the nonlinear tire model accounts for road friction changes when the tire operates in the nonlinear range.

Despite many efforts to enhance or incorporate tire models into the SSA estimation problem, these models still have limitations and can potentially contain biases. The authors of [13,14] employed different methods and models to determine the optimal solution.

Even though Gaussian filtering is a useful technique to perform state estimation, it does not work well when the model is highly nonlinear, and the posterior distribution is significantly non-Gaussian. Hence, the particle filter (PF) gained prominence in the state estimation research domain [15,16] due to its ability to guarantee accurate state estimation in the face of perturbations and the availability of supercomputers in today’s embedded systems. While [17] has proven effective in estimating SSA within the linear range of tire operation through PF, our work takes it a step further by proposing a PF for estimating SSA in both linear and nonlinear regions of tire operation, even when the vehicle experiences significant variations in lateral acceleration. Moreover, our models also consider significant fluctuations in the vehicle’s lateral acceleration, leading to deviations in vehicle models from the actual vehicle.

This paper’s primary contribution is the provision of both small and large SSA estimates, independent of road adherence and driving scenario, through a combination of multiple filters using a Dugoff tire model with constant cornering stiffness. Additionally, this paper highlights a cross-combination approach using a machine learning algorithm that enables the selection of the appropriate model and filtering technique based on the driving context.

This paper is organized as follows: Section 2 presents the recent works on estimating vehicle SSA. Section 3 expresses the Kalman filter protocol and its variant. Section 4 introduces the notion of particle filter along with the importance resampling technique. Section 5 describes the different motion and observation models used in this work. Section 6 is devoted to the concept and challenges of cross-combining several filters using data-driven methods. The results are presented and discussed in Section 7 and Section 8, respectively. Lastly, the conclusions and future work are presented in Section 9.

## 2. The Recent Works on Estimating Vehicle Sideslip Angle

The precision of sideslip angle (SSA) estimation is a key concern in the field of automotive control systems as it is the gauge measure of vehicle controllability. To ensure that the driving behavior remains predictable, an SCL or TV algorithm requires a sensed SSA value to surpass a critical threshold of SSA. To this end, refs. [18,19,20] have developed data-driven SSA estimation algorithms giving satisfactory results even in the presence of complex dynamics and uncertain road conditions. However, refs. [21,22,23] caution against overreliance on these methods, as their accuracy is heavily dependent on the quantity and quality of data used in the training process of the neural nets, which is not always guaranteed since one cannot cover all the driving scenarios in an exhaustive way. Moreover, refs. [24,25,26] assert that these approaches also suffer from the problems of embeddability and the explicability of neural networks, while bearing in mind that even in the presence of a large amount data sets, the study of uncertainty on the prediction quality of these neural networks is still to be demonstrated.

To tackle these challenges, the authors of [27,28,29] have proposed a hybrid method combining classical model-based estimation methods with learning as already investigated in [30,31].

The goal is to leverage the knowledge of the hidden dynamics of the vehicle since neural nets offer the opportunity to learn the relationship between sensor measurements and the sideslip angle through data analysis. Examples of data-driven approaches include artificial neural networks (ANNs) [32], support vector machines (SVMs) [19,33], and random forests [34]. Thus, the estimation errors of the filter are corrected, for example, by learning the gain of the observer.

Robustness of the estimation algorithm in the presence of perturbations and model uncertainties is also a critical consideration. Although [35,36] have demonstrated the robustness of their solution in various driving scenarios, most algorithms presented in the literature are limited to estimating SSAs below 10°, apart from [37], which goes beyond that. To this end, refs. [38,39,40] promote the estimation of both the SSA and the road friction coefficient allowing for the adaptation of the estimated SSA signal to the ground truth. In a similar vein, refs. [41,42] employ nonlinear tire models with an adaptation protocol on the value of the tire stiffness coefficient to reflect the adhesion levels at the tire–ground interface. Ref. [43] proposes a Kalman filter with a covariance matrix adaptation law, allowing for the transition between linear and nonlinear tire models for reliable SSA estimation regardless of the tire behavior. In contrast, refs. [44,45,46] proposed approaches that do not require the knowledge of the grip conditions but only use the inertial measurement data. The results are favorable even in the presence of strong lateral acceleration variation occurring in the linear zone [47] of the tire operation. On the other hand, other nonlinear model-based filtering methods such as fuzzy logic [48], sliding mode [49], or H- infinite [50] have also been applied to SSA estimation, as in [51,52,53]. In 2018, the authors of [54] formulated the estimation problem as an optimization problem using the gradient descent method. The cost function was designed using the nonlinear tire model, the magic formula [55] and the gradient formula. The results show that the method can accurately and reliably estimate the vehicle’s SSA and road friction coefficient under different test conditions. With the advent of more accurate sensing devices on vehicles, new approaches to SSA estimation, known as vision-based observers, have emerged. These approaches use cameras and image processing algorithms to estimate the sideslip angle [56,57,58]. They require visual features such as road markings or landmarks to be visible in the camera images, without which the estimation may result in failure. However, the transmission of visual information from on-board cameras to the filter may introduce delay and a low sampling rate; this issue was addressed by adding a delay compensator, which resulted in a decrease in the mean square error of estimation by a factor between 2 and 10, depending on the type of maneuver.

Sensor fusion [59,60] methods merge data from several sensors to estimate the SSA. These methods typically use a combination of sensors such as accelerometers, gyroscopes, GPS, and vision sensors to provide more accurate estimates. A common approach involves a Kalman filter [61,62], which uses a combination of measurements from sensors and a vehicle dynamics model to estimate the SSA. An interacting multiple model (IMM) Kalman filter combining two extended Kalman filters (EKFs), each incorporating kinematic and dynamic equations related to the vehicle’s lateral velocity, is investigated in [63]. This method estimates vehicle SSA by leveraging sensor fusion using both in-vehicle sensors and a low-cost standalone global positioning system (GPS). In a similar vein, the authors of [64] proposed a double Kalman filter (DKF) for state and parameters estimation, where two estimation stages are based on cascade stability theory in the continuous time domain. The first stage ensures global convergence, while the second stage compensates for the potential loss in performance by utilizing the estimate obtained from the first stage through local linearization techniques. The performance of a two-stage update filter was demonstrated by [65], where an ensemble Kalman–Bucy filter for the continuous time filtering problem is used concurrently with a generalized ensemble transform particle filter for intermittent parameter updates. Further, to improve the accuracy of error covariance matrix estimation in the uncentered Kalman filter (UKF), ref. [66] presented a mixed Kalman filter (MKF) for passive radar target tracking models with significantly enhanced performance. Finally, the results in [67] revealed that multiple model (MM) filters provide more reliable estimations by employing multiple filters with different models running in parallel and the outputs of each filter are fused by assigning probabilities to each filter’s estimations, either using multiple model adaptive estimation (MMAE) or the interacting multiple model (IMM).

## 3. The Kalman Filtering Approach

### 3.1. The Kalman Filter

The Kalman filter belongs to the family of the Bayesian filters proposed by Rudolf E. Kalman, answering the question: “how to estimate the state of a system supposed linear assuming a Gaussian distribution”. The KF estimates a state x $\in R^{n}$ of a discrete time-controlled process that is governed by the linear stochastic difference equation:(1)$${\hat{x}}_{k|k-1}={A_{k-1}\hat{x}}_{k-1|k-1}+B_{k-1}u_{k-1}+q_{k-1},$$
with a measurement $y\in R^{m}$ that is:(2)$$y_{k}=H_{k}x_{k}+r_{k},$$

The matrices *A*, *B*, and *H* represent, respectively, the state matrix, the input matrix related to the actuation, and the output matrix. The process noise $q_{k}$ and measurement noise $r_{k}$ are random variables independent of each other and considered Gaussian. In practice, the covariance of the measurement and process noise vary with each time instant or measurement. Here, we assume both are constant. The Kalman filter aims at minimizing an estimation error by tackling or counteracting deviation between the a *posteriori* state (corrected state after prediction) and the a *priori* state (predicted state according to the plant model) called estimation error.

The newly estimated state is a linear combination of an estimate of the a *priori* state and a weighted difference between the real measurement that is the ground truth and the predicted measurement, as described by the equation:(3)$${\hat{x}}_{k|k}={\hat{x}}_{k|k-1}+K_{k}\left(y_{k}-H_{k}{\hat{x}}_{k|k-1}\right)$$
with $K_{k}$ the Kalman gain, $y_{k}$ the measurement at time k, and ${\hat{x}}_{k|k}$ the current state estimate. As for all Bayesian filters, a KF algorithm involves a prediction phase where the state transition function is used to forecast the evolution of the system over time:(4)$${\hat{x}}_{k|k-1}={A_{k-1}\hat{x}}_{k-1|k-1}+B_{k-1}u_{k-1}$$
(5)$$P_{k|k-1}=A_{k-1}P_{k-1|k-1}A_{k-1}^{T}+Q_{k-1}$$
where ${\hat{x}}_{k|k-1}$ indicates the prior state estimate at time step k given the previous state estimate at time k−1, and $P_{k|k-1}$ is the covariance (the probability density) associated to that state.

The prediction phase is followed by the update phase where the estimate (the prior) and its covariance are updated based on the following set of equations:(6)$$K_{k}=P_{k|k-1}{H^{\prime }\left({\hat{x}}_{k|k-1}\right)}^{T}{S_{k}}^{-1}$$
(7)$$S_{k}=H^{\prime }\left({\hat{x}}_{k|k-1}\right)P_{k|k-1}{H^{\prime }\left({\hat{x}}_{k|k-1}\right)}^{T}+R_{k}$$
(8)$${\hat{x}}_{k|k}={\hat{x}}_{k|k-1}+K_{k}\left(y_{k}-H{(\hat{x}}_{k|k-1})\right)$$

The Kalman filter is computationally efficient and provides an optimal estimate of the state variables for linear and Gaussian systems. A complete picture of a KF combining both the prediction and the update steps is illustrated by Figure 1.

### 3.2. The Extended Kalman Filter

As mentioned above in Section 3.1, the KF addresses the general problem of estimating the state x $\in R^{n}$ of a discrete time-controlled process that is governed by the linear stochastic difference equation. Alternatively, the extended Kalman filter computes the state estimate of a process in the face of nonlinearities by applying the KF protocol to the linearized system around the current estimate using the partial derivatives of the state transition and observation functions.

The parameters involved in the mechanism of an EKF, as described in Figure 2, are:

$A_{\left[i,j\right]},$ is the Jacobian matrix of partial derivations of f with respect to x, that is,
$$f^{\prime }\left({\hat{x}}_{k-1|k-1}\right)=\frac{\partial f_{\left[i\right]}}{\partial x_{\left[j\right]}}\left({\hat{x}}_{k-1|k-1},u_{k-1},0\right)=A_{\left[i,j\right]}$$

$Q_{\left[i,j\right]},$ is the Jacobian matrix of partial derivations of f with respect to q,
$$Q^{\prime }\left({\hat{x}}_{k-1|k-1}\right)=\frac{\partial f_{\left[i\right]}}{\partial q_{\left[j\right]}}\left({\hat{x}}_{k-1|k-1},u_{k-1},0\right)=Q_{\left[i,j\right]}$$

$H_{\left[i,j\right]}$, is the Jacobian matrix of partial derivations of de h with respect to x,
$$H^{\prime }\left({\hat{x}}_{k-1|k-1}\right)=\frac{\partial h_{\left[i\right]}}{\partial x_{\left[j\right]}}\left({\hat{x}}_{k},0\right)=H_{\left[i,j\right]}$$

$R_{\left[i,j\right]},$ is the Jacobian matrix of partial derivations of h with respect to r,
$$R^{\prime }\left({\hat{x}}_{k-1|k-1}\right)=\frac{\partial h_{\left[i\right]}}{\partial r_{\left[j\right]}}\left({\hat{x}}_{k},0\right)=R_{\left[i,j\right]}$$

The plant model is discretized according to the Euler method.

## 4. The Particle Filtering Approach

### 4.1. The Particle Filtering Methodology

Bayesian filtering is a suitable technique to perform the state estimation of a linear system. However, it is not efficient when the system’s models are strongly nonlinear or when the a posteriori distribution is significantly non-Gaussian.

The particle filter provides an alternative to the state estimation of a strongly nonlinear and non-Gaussian system. Particle filtering is a recursive Monte Carlo method based on the use of nonparametric representations (mean and covariance) to approximate the a posteriori density $P\left(x_{k}|y_{1:k}\right)$ over time by a set of N random samples called particles, where $y_{1:k}=\left\{y_{1},\cdots ,y_{k}\right\}$ is a set of observations stored up to *k*. The filtering density is calculated as a weighted Dirac sums,
(9)$$P\left(x_{k}|y_{1:k}\right)\approx \sum_{i=1}^{N}\omega_{k}^{i}*\delta \left(x_{k}-x_{k}^{i}\right).$$
where $x_{k}^{i}$ are particles and $\omega_{k}^{i}$ are associated weights. Each particle represents one possible state realization and its associated weight represents how probable that state realization is.

The mechanism of a particle filter algorithm performs (or consists of) two major steps. First, the prediction step, where the particles drawn from a known distribution q(x) are propagated as follows,
(10)$$x_{0}^{i}\;\sim \;q\left(x_{0}\right),\;\;with\;\;\omega_{0}^{i}=\frac{1}{N}$$
(11)$$x_{k}^{i}\;\sim \;q\left(x_{k}^{i}|x_{k-1}^{i},y_{k}\right),$$
according to the motion model f and the observation model h given by
(12)$$\left\{\begin{array}{c} x_{k}=f\left(x_{k-1},u_{k-1},v_{k}\right)\\ y_{k}=h\left(x_{k},u_{k},w_{k}\right). \end{array}\right.$$

Each particle is defined by a set of parameters ${\left\{x_{0:k}^{i},\omega_{k}^{i}\right\}}_{i=1}^{N}$ where N, $q\left(x_{0}\right),$ $x_{0}^{i}$, and $\omega_{0}^{i}$ are, respectively, the number of particles, the initial state distribution, the initial particle generation, and their associated weights. The particles are propagated ahead according to a uniform and non-Gaussian distribution, with $v_{k}$ as the process noise and $w_{k}$ as the measurement noise which are considered non-Gaussian. That is, at the initialization stage all particles have equal probability density.

q(x) is a known distribution called importance density or the proposal density from which samples, $x^{\left(1\right)},x^{\left(2\right)},\cdots ,x^{\left(N\right)}$, are generated, since it is difficult to sample from p(x) the supposed unknown. Therefore, q(x) is equivalent to p(x) such that one over the other is always one. The second step involves computing the state estimate, the computation, and the normalization of the weights affected to each particle at each time instant k condition on the previous weights. That is, the new weight is set proportional to the previous weight scaled by the likelihood of the particles $P\left(y_{k}|x_{k}^{i}\right)$ given the data and how probable it is according to the motion model $P\left(x_{k}^{i}|x_{k-1}^{i}\right)$ and normalized by the $q\left(x_{k}^{i}|x_{k-1}^{i},y_{k}\right)$ proposal density, expressed as follows:(13)$${\tilde{\omega }}_{k}^{i}\propto \omega_{k-1}^{i}\frac{P\left(y_{k}|x_{k}^{i}\right)P\left(x_{k}^{i}|x_{k-1}^{i}\right)}{q\left(x_{k}^{i}|x_{k-1}^{i},y_{k}\right)}.$$

Afterwards, to transition from state k−1 to state k, the weights are normalized such that they sum to one through the following expression:(14)$$\omega_{k}^{i}=\frac{{\tilde{\omega }}_{k}^{i}}{\sum_{i=1}^{N}{\tilde{\omega }}_{k}^{i}}$$

The two probability densities at the heart of the operation of the particle filter, resulting from Equation (13), are the *prior density* $P\left(\left.x_{k}^{i}\right|x_{k-1}^{i}\right)$ and the *likelihood* $P\left(\left.y_{k}\right|x_{k}^{i}\right).$ The former is derived using the state transition function condition on the previous state distribution; the latter is approximated based on the observations in accordance with the following equation:(15)$$L(x)=\frac{1}{{\left(2\pi \right)}^{N/2}{\det\Sigma }^{1/2}}\exp\left[\left.-\frac{1}{2}{\left(x-\mu \right)}^{T}\Sigma^{-1}(x-\mu )\right]\right.$$

In Equation (15), *N* represents the number of state variables and *μ* is a vector of means associated with the state vector x and with a variance Σ. The last part of the update step consists of deriving the best state estimate at each time step as the weighted sum of the particles:(16)$${\hat{x}}_{k}=\sum_{i=1}^{N}\omega_{k}^{i}x_{k}^{i},$$

The particle with the greatest weight has a larger impact on the state estimate and the higher the number of particles, the better the convergence, provided that we have a high computation power.

### 4.2. The Particle Filter Modus Operandi

In the actual implementation of the particle filter it is necessary to define four important parameters which are the sample time, the process noise covariance, the measurement noise covariance, and the initial distribution.

Our proposed PF scheme includes the extended Kalman filter SSA estimation as an additional input to the particle filtering problem, alongside the four ingredients listed above. The EKF state estimate is leveraged to enhance the final PF estimation. By comparing the observed balance between the EKF estimate and the PF estimate, represented as
(17)$$∆\varepsilon_{SSA}=\left|{SSA}_{EKF}-{SSA}_{PF}\right|,$$
we can then update the PF weights. Hence, this deviation serves as a pseudo-measure for the PF and allows for the accurate computation of the likelihood function. In other words, this deviation enables a more accurate recalculation of the likelihood function.

In our study, several samples of experimental data are preprocessed offline to determine the sample time and the covariances prior to the operation of the filter. The particles are spread out following a uniform and non-Gaussian distribution, as mentioned earlier. Like most of the model-based estimation methods, the concept of particle filtering work modulo the correct modelling of the state transition function and the measurement model. However, during the full operation of a particle filter algorithm as illustrated by Figure 3, if the covariances are well-tuned and the motion model is designed in such a manner that it closely captures the dynamic of the system, the only compromise that may be struck to obtain the best guess would be the selection of the number of particles.

In contrast with the previous example, if we obtain a poor estimate irrespective of the above-mentioned conditions, it implies that one of the particles is associated with a very high weight, whereas the rest of the particles have a weight close to zero. It means the set of particles generated are no longer contributing to the description of our posterior anymore. Thus, most of our particles are no longer representative of the systems’ evolution. This phenomenon is known as the particle’s degeneracy. To remedy such a situation, resampling is required. The idea of resampling is to use Monte Carlo sampling to generate a better description of our filtering density. Basically, new particles are re-generated by drawing independent samples from our current posterior to replace the old sample set with the new one and set all weights to 1/N. Resampling requires some extra calculations and may introduce some errors but improves performance immensely over time.

## 5. The Motion and Observation Models of the Vehicle

A comprehensive system model is required in any model-based design approach for system analysis and algorithm development. From an observer design perspective, the model should ideally capture the fundamental dynamics while remaining simple enough to serve as the basis for model-based observers. On the flip side, the model must also have sufficient fidelity to allow performance evaluation in simulation, thereby reducing the risks and costs associated with experimental validation.

For this purpose, we have modeled various models to capture the fundamental dynamics that play a role in the behavior of the prototype vehicle. The tests performed on the Alpine A110 are shown in Figure 4. It is important to note that all tests are carried out on snow (see Figure 4).

### 5.1. The Kinematic Vehicle Model

The so-called kinematic model of the vehicle is described by the empirical equations:(18)$$a_{x}=\eta =v_{y}*\left(\frac{{\dot{v}}_{x}}{v_{y}}-\dot{\psi }\right)=\left({\dot{v}}_{x}-\dot{\psi }{.v}_{y}\right)$$
(19)$$a_{y}=\gamma =v_{x}*\left(\dot{\beta }+\dot{\psi }\right)=\left({\dot{v}}_{y}+\dot{\psi }{.v}_{x}\right).$$

The state space representation of the model (the kinematic model) of the system to be observed is written as follows:(20)$$\frac{d\left[\begin{array}{c} v_{x}\\ v_{y} \end{array}\right]}{dt}=\left[\begin{array}{cc} 0 & \dot{\psi }\\ -\dot{\psi } & 0 \end{array}\right]\left[\begin{array}{c} v_{x}\\ v_{y} \end{array}\right]+\left[\begin{array}{cc} 1 & 0\\ 0 & 1 \end{array}\right]\left[\begin{array}{c} a_{x}\\ a_{y} \end{array}\right]$$
(21)$$y=Cx=\left[\begin{array}{cc} 1 & 0 \end{array}\right]\left[\begin{array}{c} V_{x}\\ V_{y} \end{array}\right],$$
with $V_{x}$ as the vehicle velocity, $a_{x}$ as the acceleration along the longitudinal axis, $V_{y}$ as the lateral velocity of the vehicle, $a_{y}$ as the lateral acceleration, and $\dot{\psi }$ or r as the yaw rate.

The kinematic model does not explicitly involve the lateral forces of the axles, thus reflecting a strong assumption about the tire model chosen in a vehicle dynamic behavior analysis. The emphasis of this model lies in its use of easily accessible quantities, provided by sensors found on all vehicles.

It is also required to have an observation model relating the measurement to the state. In this application, $V_{x}$ is measurable and the observation model is described by Equation (21). The limitations of the kinematic model are important to know.

First, the model is only observable for nonzero values of $\dot{\psi }$. When the yaw rate is zero or close to zero, it is very probable that the vehicle is driving in a straight line. In this scenario, SSA control logic is disabled and SSA estimation is not required. Second, the model assumes that there is no slope and that the tire–road contact is always slip-free. Therefore, in some cases, a more complete model should be considered which considers other intrinsic aspects of the car as well as its dynamics that the kinematic model does consider.

### 5.2. The Dynamic Vehicle Model

The nonlinear dynamic model is based on the fundamental principle of dynamics; the accelerations at the center of gravity and the forces at the tires are expressed (see Appendix A and Appendix A). The complete four-wheel model can be simplified into a two-wheel model called “bicycle model” where we consider a symmetry along the longitudinal axis of the car and no rear steering. The nonlinear bicycle model is expressed as,
(22)$$\frac{d\left[\begin{array}{c} \dot{\psi }\\ \beta \end{array}\right]}{dt}=\left\{\begin{array}{c} \ddot{\psi }=\frac{a_{1}(F_{y_{f}}\cos\delta +F_{x_{f}}\sin\delta )+a_{2}F_{y_{r}}\sin\delta }{I_{z}}\\ \dot{\beta }=\frac{F_{y_{f}}\cos\left(\delta -\beta \right)+F_{y_{r}}\sin\delta +F_{x_{f}}\sin\left(\delta -\beta \right)}{mv_{x}}-\dot{\psi } \end{array}\right.$$
and the observation equations are,
(23)$$y\left(\left[\begin{array}{c} \psi \\ a_{x}\\ a_{y} \end{array}\right]\right)=\left\{\begin{array}{c} \dot{\psi }=\dot{\psi }\\ a_{x}=\frac{1}{m}\left(F_{x_{f}}\cos\delta +F_{x_{r}}-F_{y_{f}}\sin\delta \right)\\ a_{y}=\frac{1}{m}\left(F_{x_{f}}\sin\delta +F_{y_{r}}+F_{y_{f}}\cos\delta \right) \end{array}\right.$$

To express the front and rear axle longitudinal forces $F_{x_{f}},F_{x_{r}}$ and front and rear axle lateral forces $F_{y_{f}},F_{y_{r}},$ different tire models are available. One of the reference models is the model of Hans B. Pacejka’s, also called “magic formula”. However, it requires a thorough knowledge of the tire parameters and road conditions. A more basic linear model is therefore used, which relates the force to the front and rear axles’ sideslip angle $\beta_{f,r}$ such that:(24)$$\alpha_{f,r}=\delta_{f,r}-\beta_{f,r}$$
(25)$$F_{y_{f},y_{r}}=C_{\alpha f,\alpha r}\alpha_{f,r}$$
with β as the vehicle SSA, $C_{f,r}$ as the front and rear axle tires cornering stiffness coefficient, $\delta_{f,r}$ as the front and rear axle steer angle, and $\alpha_{f,r}$ as the tire slip angle defined in Figure 5. Using the small angle approximation, the bicycle reads as:(26)$$\dot{\beta }=-\beta \frac{C_{\alpha f}+C_{\alpha r}}{mv_{x}}-\dot{\psi }\left(1+\frac{a_{1}C_{\alpha f}-a_{2}C_{\alpha r}}{m{v_{x}}^{2}}\right)+\delta_{f}\frac{C_{\alpha f}}{mv_{x}}$$
(27)$$\ddot{\psi }=-\beta \frac{a_{1}C_{\alpha f}-a_{2}C_{\alpha r}}{I_{z}}-\dot{\psi }\frac{{a_{1}}^{2}C_{\alpha f}-{a_{2}}^{2}C_{\alpha r}}{I_{z}v_{x}}+\delta_{f}\frac{a_{1}C_{\alpha f}}{I_{z}}$$
(28)$$a_{y}=-\beta \frac{C_{\alpha f}+C_{\alpha r}}{m}+\dot{\psi }\frac{a_{2}C_{\alpha r}-a_{1}C_{\alpha f}}{mv_{x}}+\delta_{f}\frac{C_{\alpha f}}{m}$$
where $a_{1}$ and $a_{2}$ are the lengths outlined in Figure 5 and $I_{z}$ and m are the vehicle inertia and vehicle mass, respectively. The SSA is directly estimated in the state vector. As mentioned before, the assumption made on the tire cornering stiffness coefficient, considered constant, can influence the SSA estimation because it varies according to the road conditions. That is why the random walk dynamic model has been studied, to compensate for the lack of knowledge of the friction coefficient, a crucial variable for an optimal SSA estimation.

### 5.3. The Random Walk Dynamic Vehicle Model

The random walk dynamic model constitutes Equation (22) augmented by the tire force dynamic. The state vector now includes six parameters ${\left[\begin{array}{cc} \begin{array}{cc} \dot{\psi } & \beta \end{array} & \begin{array}{cc} \begin{array}{cc} F_{x_{f}} & F_{x_{r}} \end{array} & \begin{array}{cc} F_{y_{f}} & F_{y_{r}} \end{array} \end{array} \end{array}\right]}^{T}$. The random walk principle supposes that the forces are constant in the state model, hence their temporal derivatives are null. Additionally, a noise is added to the filtering process accounting for random force variations if they are not constant. The state transition function is:(29)$$\left\{\begin{array}{c} \ddot{\psi }=\frac{a_{1}\left(F_{y_{f}}\cos\delta +F_{x_{f}}\sin\delta \right)+a_{2}F_{y_{r}}\sin\delta }{I_{z}}\\ \dot{\beta }=\frac{F_{y_{f}}\cos\left(\delta -\beta \right)+F_{y_{r}}\sin\delta +F_{x_{f}}\sin\left(\delta -\beta \right)}{mv_{x}}-\dot{\psi }\\ {\dot{F}}_{x_{f}}={\dot{F}}_{x_{r}}={\dot{F}}_{y_{f}}={\dot{F}}_{y_{r}}=0 \end{array}\right..$$

The measurement function is equivalent to Equation (23).

This model not only exhibits prompt convergence akin to the kinematic model, but also provides an estimate of the lateral and longitudinal forces exerted on the car. This estimation can serve as a valuable input in determining the road friction coefficient.

### 5.4. The Dynamic Vehicle Model Validation

The validation of the model is completed with an empirical calculation. The parameters of the Alpine A110, the steering wheel angle input by the driver and the speed of the vehicle allow for the calculation of the different state variables according to the dynamic model.

Figure 6 and Figure 7 attest that the model design matches the target vehicle well, although the lateral acceleration (see Figure 8) does not match as perfectly as the SSA, and the yaw rate is not far off either.

These signals from the calculation are then compared with the corresponding signals from the sensors.

The lateral acceleration displayed in Figure 8 exhibits a marked variation up to the 0.6 g level. It is important to note that these measurements were obtained during tests conducted in Sweden on snow-covered tracks, indicating a friction coefficient of approximately 0.3. As stated in [68], the lateral forces on a vehicle’s rolling axles must satisfy the following constraint:(30)$$\left|a_{y}\right|=\left|\frac{V_{x}^{2}}{R}\right|=\left|\dot{\psi }*V_{x}\right|\leq \mu_{max}g$$
where R is the radius of turn, g is the gravitational constant, and $\mu_{max}$ is the maximum road adhesion coefficient.

The lateral acceleration a vehicle can develop is constrained by the maximum lateral friction coefficient at the tire–road interface [69]. In our example, the lateral acceleration surpasses this constraint, indicating a significant nonlinearity (0.6 g≥0.3 g) that must be coped with by our estimator.

## 6. Data-Driven Cross-Combination Technique: A Variable Structure Observer Scheme

### 6.1. The Cross-Combination: Definition, Concept, and General Detail Functional Diagram

The core concept of data-driven cross-combination is the use of data insights to classify the various operation regimes of a system (vehicle) for the purpose of switching between two or more observers (see Figure 9), thereby forcing the reconstruction of the system states towards a preferred solution by design. This leads to a variable structure observer (VSO) that can reduce the impact of uncertainties, disturbances, and nonlinear dynamics on the state estimation problem.

As depicted in Figure 9, a set of estimators, carefully selected by the control engineer, work concurrently. These estimators are fed with measurement data from both the system’s (vehicle) sensor devices and the control inputs proceeding from the driver’s inceptors.

The optimal solution or preferred solution is influenced by both the selected machine learning (ML) model (as shown in Figure 10) and well-established analytical metrics, such as the root means square error and maximum error. Finally, the cross-combination algorithm (CCA) computes an optimal solution by considering the states estimated by each individual filter.

### 6.2. The Cross-Combination Paradigm and Methodology for Vehicle SSA Estimation

Every vehicle estimator design is about the art of compromise since each estimation method offers benefits and drawbacks. Using the cross-combination method for a robust sideslip angle estimation is (amount to) taking the benefit of two different filtering approaches or more and getting rid of their penalties, resulting in a preferred estimation method depending on the vehicle’s operating point or desired performance gain.

Our study involves cross-combining three filtering methods, selecting the appropriate motion model and filter for the best estimate of the sideslip angle over time, in a transient or steady-state vehicle operating regime.

This approach combines the benefits of different process models to achieve accurate and robust estimation regardless of the driving situation. Several specific cases can be identified:

When the value of the lateral or longitudinal velocity is close to zero. In this case, the kinematic model does not allow for good estimation of the sideslip angle. It is thus preferable, for operating regions where the value of the speed is close to zero, to use the dynamic model.When the yaw rate is close to zero. The matrix $\left[\begin{array}{cc} 0 & \dot{\psi }\\ -\dot{\psi } & 0 \end{array}\right]$ is no longer observable. Therefore, the kinematic model cannot estimate the sideslip angle in this case. One of the dynamic models can then be used.Increasing the number of particles propagated in a particle filter can influence the efficiency of a model to estimate the sideslip angle. On the other hand, increasing the number of particles is not a viable solution given the limitation of the computing power available in a vehicle. It is ingenious to use the particle filter only in the case where the two other filters can no longer yield a good estimate of the sideslip angle.The choice of model and filter depends mainly on the tolerated error.

The cross-combination is performed using a machine learning algorithm that receives data from the various vehicle’s sensor devices, like accelerometer and gyroscope, over ten samples every 200 ms. These data are used in the training phase to predict the vehicle’s operating regime. Our filter selection for the initial implementation of cross-combination is motivated by the specific characteristics of the vehicle’s yaw dynamics, which include a significant degree of nonlinearity, caused by the unpredictable irregularity (fluctuations) in the road adhesion level at the tire–road interface. Additionally, we considered the impact of the signal measurement noise from various sensors and the available computational resources.

This is why we have chosen the KF which is computationally efficient but limited to linear and Gaussian systems, and the PF which allows estimating multiple modes of the posterior distribution and to handle situations where the system is highly nonlinear, or the noise is non-Gaussian.

As depicted in Figure 11, each filter proposes a state estimate at each time step and our algorithm disposes of a preferred solution and subsequently sets to zero any other estimate output, similar to a combinational logic framework.

The machine learning algorithm, having learned from a mass of driving data, can distinguish the operating regime and the vehicle model adapted to this regime, and then decides, thanks to a decision tree according to a combinational logic framework, which model and filter to select for the estimation of the sideslip angle.

### 6.3. Data Set Labeling

Prior to determining the structure of the machine learning algorithm to be used, the choice of the estimation model was transformed into a classical supervised learning problem. Given a sample of data,
(31)$${\left(x_{k},y_{k}\right)}_{1\leq k\leq n},$$
with $x_{k}$ as the state vector to be estimated, $y_{k}$ as a label representing the model adapted to each time k, and n as the number of samples. This formulation reflects a classification problem, which can be illustrated by a function:(32)$$f_{n}:x\mapsto y\in Y$$
where the model shall be able to predict the values of y∈Y associated to each allowable value of x∈X.

In the context of this paper, y is a Boolean, where 0 denotes that the model estimate will not be selected, and 1 that the model estimate will be selected. On a dataset of $n_{s}=10$ samples, the root mean square (RMSE) of the sideslip angle estimate is calculated for each model:(33)$$\epsilon =\sqrt{\frac{1}{n_{s}}\sum_{k=1}^{n_{s}}{\left({\hat{\beta }}_{k}-\beta_{k}\right)}^{2}}$$

Then the model with the lowest RMSE will be selected for the sideslip angle estimation. So, during the training, the algorithm learns which model is the most accurate based on the lateral acceleration and yaw rate data fed into the ML algorithm. This allows for the nonlinearities and road conditions within the estimation process to be taken into account. In the scope of this investigation, several datasets are used for learning, and the sideslip angle sensor measured data serve as ground truth.

### 6.4. The Training and Validation Data Sets Details

The dataset used to train and validate our model was collected at the Arjeplog Test Management center in Sweden, using an Alpine A110 car and the newly released Renault Austral shown in Figure 12. Both vehicles were equipped with a range of high precision sensing devices, including wheel speed sensors, yaw velocity sensors, and steering wheel angle sensors for both front and rear wheel steering systems, as well as GPS, gyroscope, accelerometer, vehicle sideslip sensor, suspensions travel sensor, brake pressure sensors, and more. This resulted in a comprehensive dataset that included all signals transmitted by the vehicle’s sensors and ESC block, such as wheel braking pressure, engine torque, SSA, vehicle longitudinal and lateral acceleration, vehicle forward and lateral speed, yaw velocity, wheel speeds, and rear wheel steer angle. The dataset covered a range of vehicle dynamics evaluation maneuvers, including the circle (at constant radius and speed), U-turn, J-turn, single and double lane change, slalom, pulse steer, close circuits, combined traction and braking phases, and front and rear steering systems triggered both separately and simultaneously.

To ensure the model’s performance was not overly dependent on a specific subset of the data, we utilized standard cross-validation techniques on five folds. The datasets of both cars were partitioned into two subsets, where 75% was used for training the model, 15% for validation, and 10% for testing the model’s performance. The signals were recorded through CANalizer at a frequency of 100 Hz. We will provide a link giving readers access to the datasets in the reference section. The test data which are plotted in the Results Section only consist of the dataset relating to the Alpine 110.

The training and validation data entailed all the handling maneuvers cited above for both cars. The test set, on the other hand, included a full driving trajectory of the Alpine A110 car.

### 6.5. The Decision Tree Mechanism

A decision tree is composed of two types of nodes: decision nodes associated with conditions and leaves associated with labels. The tree is derived from the training data. It partitions the training data space into subpartitions that predict the best label. The maximum and minimum lateral acceleration and yaw rate values are filled in.

The entropy is the metric that allows us to determine the level of homogeneity of the data associated to a partition:(34)$$E=-\sum_{i=1}^{N}p_{i}\log_{2}p_{i},$$
where N is the number of classes =3 and $p_{i}$ is the probability of randomly drawing an element class i. The entropy is expressed as:(35)$$E=-\left(p_{k}\log_{2}p_{k}+p_{d}\log_{2}p_{d}+p_{p}\log_{2}p_{p}\right)$$
where $p_{k}$ is the probability of deriving an acceleration/yaw rate component consistent with a kinematic model estimation, $p_{d}$ is the probability associated with a dynamic model estimate, and $p_{p}$ is the probability associated with a bicycle model estimate via the particle filter method.

The gain or information reflects the quality of the choice of a threshold or subpartition (or son) under a parent partition (father node):(36)$$Gain=E_{father}-E_{son}$$

Each threshold in the tree will have a gain assigned to it, which will provide information about its relevance, i.e., its ability to reduce entropy, thereby increasing the gain for the lower node.

The confusion matrix (a confusion matrix is a table that is used to evaluate the performance of a classification algorithm; it is used in machine learning to measure how well a model is able to predict different classes of a given dataset) results indicate that 844 items classified in the kinematic estimators belong to this class and 1152 samples classified in the dynamic estimators belong to this class.

## 7. Results

### 7.1. Results with the Kalman Filter Using the Kinematic Vehicle Model (Experimental Data)

The results achieved in this section are compared to the test data recorded during a complete circuit driving scenario on a snowy track at the ATM test center in Sweden. The state estimators are initially assessed through simulation data, then followed by real data recorded on the Alpine A110 vehicle. Since the simulation results regarding the longitudinal velocity are conclusive because it is our measurement model, only the lateral velocity state estimate is presented. The state vector to be estimated is:(37)$${\left[\begin{array}{cc} V_{x} & V_{y} \end{array}\right]}^{T},$$
where the vehicle forward speed contributes to the estimation of the lateral speed. The kinematic model’s capability to estimate the lateral velocity, $V_{y}$, using Kalman filtering is supported by the findings presented Figure 13. The results reveal a maximum error of 1.5 m/s between the measured and estimated lateral velocities, demonstrating the model’s efficacy.

Once the two velocity components (longitudinal and lateral velocity) are estimated, the SSA estimate is obtained using (small angle approximation):(38)$$\beta =arctan\left(\frac{v_{y}}{v_{x}}\right),$$

It is noteworthy that, during the initial 0 to 1.8 s time interval, the longitudinal velocity of the vehicle remains near zero while the lateral velocity remains constant. This can be attributed to the arctangent of infinity with respect to Equation (38), which is equivalent to pi/2 or 1.57 and corresponds to the amplitude of the peak or jump observed at the start of Figure 14. (Division by a small number would imply a very large number.
(39)$$\underset{x\to \infty }{\lim}arctan(x)=\frac{\pi }{2}⟺\underset{\frac{v_{y}}{v_{x}}\to \infty }{\lim}arctan\left(\frac{v_{y}}{0}\right)\approx 1.57$$

Such observations are a direct consequence of the extremely low or zero longitudinal velocity during the initial moments of the vehicle’s movement.

Figure 13 and Figure 14 provide evidence for the effectiveness of the kinematic KF in estimating the SSA using the kinematic vehicle model. The dynamic of the estimation error illustrated by Figure 15 converges in the vicinity of zero as expected. The key performance index will be provided below.

### 7.2. Results with the Particle Filter Using the Kinematic Vehicle Model (Simulation Data)

The particle filter using the kinematic model was initially run on simulation data; the results obtained for 1000 particles and, subsequently, for 10,000 particles were compared with the measured SSA value (see Figure 16).

Figure 17 provides us with valuable insights into the impact of particle propagation on the performance of the particle filter. It is evident that the number of particles propagated significantly affects the accuracy of SSA estimation. When dealing with normal driving conditions characterized by low SSA and acceleration variations, 1000 particles are sufficient to achieve reliable SSA estimation. However, in situations such as the loss of an axle’s control (when a vehicle breaks out of control) due to excessive wheel slip, as illustrated in Figure 16, where the rear axle goes out of control around 50 to 60 s, 10,000 particles are required to maintain the same level of performance.

### 7.3. Results with the Particle Filter Using the Dynamic Vehicle Model (Experimental Data)

In Figure 18, we can observe that, by increasing the number of particles from 1000 to 30,000, a very strong performance of the particle filtering is obtained, regardless of the nonlinearities and abrupt variations in accelerations. Meanwhile, Figure 18 also reveals that the disparity between the two curves varies across different regions, indicating that a trade-off can be made between performance and computational efficiency. Therefore, it is worthwhile to explore solutions that involve adapting the number of particles based on specific criteria.

### 7.4. Results with the Particle Filter Using the Kinematic Vehicle Model (Experimental Data) and Comparison with the KF

From the analysis of Figure 19, it appears that the PF is much more accurate in terms of tracking. Nevertheless, both estimators have an acceptable performance from a chassis control point of view. Even though the error signals in Figure 20 overlap, the estimation error of the PF is much smaller than that of the KF.

It is critical to compare the performance of the particle filter with that of a simpler (more straightforward), less computationally expensive filter. The SSA estimated with the particle filter and used for comparison is the one obtained via the dynamic model. This is compared to a Kalman filter with a kinematic model.

### 7.5. Results with the Cross-Combination of Three Different Filters Using Both the Dynamic and Kinematic Vehicle Models (Experimental Data)

The results of the cross-combination of the three filtering algorithms are obtained with a smooth transition period representing the switch modes of the variable structure observer. The switching frequency needs to be in sync with the sampling frequency for embeddability purposes. Notably, the jump observed in Figure 21 and Figure 22 is because the cross-combination algorithm assigns control authority to the KF which is fed via the kinematic model, which itself relies on the arctangent function to derive the SSA.

The cross-combination method is deemed to have achieved the most accurate SSA estimation results, as demonstrated in Figure 22 using two distinct test datasets.

Our cross-combination study underlines the importance of a variable structure observer where the state is reconstructed through selecting an optimal solution (state estimate) at each time step. By comparing the cross-combination’s signal and the measured SSA, we notice yield-conclusive results (see Figure 23 and Figure 24) in terms of tracking, despite the noisy and highly nonlinear nature of the measured lateral acceleration signal, as shown in Figure 8.

Furthermore, Figure 25 demonstrates that, despite the noisy and highly nonlinear nature of the measured lateral acceleration signal, the cross-combination SSA method remains the most suitable state estimator for vehicle SSA estimation, offering both high accuracy and attractive computational efficiency. Additionally, all estimators exhibited an acceptable tracking performance and a faster convergence rate, which are crucial for real-time implementation, but with respect to RMSE and ME, cross-combination is the best.

Even though the cross-combination outperforms the EKF, and the PF in terms of ME and RMSE, as shown in Table 1, for safety reasons the PF can be enforced as the cross-combination default choice if necessary. In summary, this study affirms the cross-combination approach as the optimal method for achieving precise SSA estimation.

## 8. Discussion

In the scientific literature, there are several methods that have demonstrated their effectiveness in estimating vehicle dynamic states including SSA. Our contribution is positioned with respect to previous research in this field as a technique for leveraging multiple estimators working concurrently, resulting in a variable structure estimator.

This approach ensures a robust estimation of the sideslip angle (SSA), which is essential for real-time applications, with the provision that there is sufficient onboard computing power available. Our study assumes constant tire cornering stiffnesses, despite variations in vertical load and ground contact surface (grip level), which are difficult to access.

Although this hypothesis is fairly broad, it can afford to be since the cornering stiffnesses varies very little compared to the vertical load. Moreover, even in the case of very low adhesion like the tests conducted in Sweden, the variations in wheel force or stiffness are perceived by our estimator as disturbances to be eliminated. The results section showcases a conclusive performance in this regard.

The trouble in estimating SSA does not arise in the linear region of the tire’s operation, where a Kalman filter would do the job, but rather when one or both rolling axles slide excessively, leading to understeer or oversteer behavior. For this reason, our future works will integrate a higher layer that detects understeer or oversteer situations before planning. The ML algorithm serving as a mode switch will be augmented with a supplementary layer identifying these phenomena in real-time, further improving the performance of the cross-combination dynamic.

## 9. Conclusions

The sideslip angle estimation was conducted for Kalman and particle filters fed with a kinematic and (or) dynamic vehicle model. Several filters with corresponding vehicle models were compared. The kinematic model was found to be limited in some driving conditions, especially for low vehicle velocities (longitudinal, lateral, or yaw). Nevertheless, the dynamic model can be used in these cases. While a Kalman filter coupled to a kinematic model was generally satisfactory in the linear region of the tires, tests were also performed for a particle filter fed with the same models, which was found to be very efficient and robust in terms of sideslip angle estimation in the face of strong nonlinearity. The particle filter is computationally expensive; it is necessary to limit its use to these specific cases. The efficiency of the particle filter was demonstrated, more particularly by increasing the number of propagated particles. The computational power of embedded systems is more and more important; it will doubtlessly be possible in the future to deploy its use for a more optimal estimation in real-time applications. The cross-combination algorithm, which is a key contribution of this work, offers the possibility to use the best model associated with the best filter according to the given driving conditions. All estimators presented in this work exhibited relatively acceptable tracking performance and faster convergence rates, which are crucial for real-time implementation, but the SSA CCA observer outperformed the KF, EKF, and PF in terms of accuracy, robustness, RMSE, and ME, respectively.

Therefore, our study reports evidence supporting the notion that using a combination of multiple observers for states and parameters yields greater benefits when compared to relying solely on an individual filter. By implementing multiple filters with distinct models running concurrently, more reliable estimations can be achieved.

Still, there are certain limitations associated with this methodology, most notably its high computational requirements, as all filters are continuously activated (running) in real-time. Moreover, the transition between filters over time can potentially introduce little fluctuations within the observation signal known as “chattering”, underscoring the difficulty of effectively synchronizing the operation of multiple filters.

It is important to acknowledge that the limitations inherent to each coordinating filter also affect the overall performance of the SSA CCA mechanism.

Overall, this study provides valuable insights into the selection and evaluation of state estimators for vehicle sideslip angle estimation, which can be useful for the development of advanced driver assistance systems and autonomous vehicles. Future research will explore the combination of more than three different estimation methods (linear or nonlinear, model-based or model-free) in the cross-combination scheme. Also, we will integrate a higher layer that detects understeer or oversteer situations before planning. Hence, the ML algorithm serving as a mode switch will be augmented with a supplementary layer identifying these phenomena in real-time, further improving the performance of the cross-combination dynamic. In conclusion, one of our future challenges would be to extend the application of this SSA estimation method across the complete spectrum of vehicles within the Renault group.

## 10. Patents

The research discussed in this article has led to the submission of a European patent application, currently under examination.

## Figures and Tables

**Figure 1 sensors-23-05877-f001:**
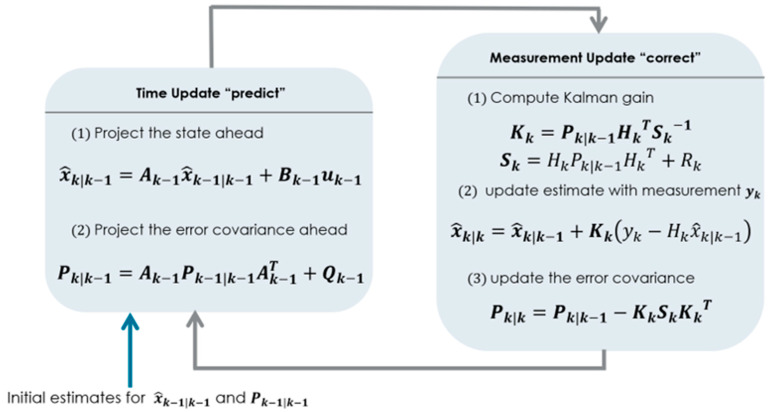
A complete picture of the operation of a Kalman filter.

**Figure 2 sensors-23-05877-f002:**
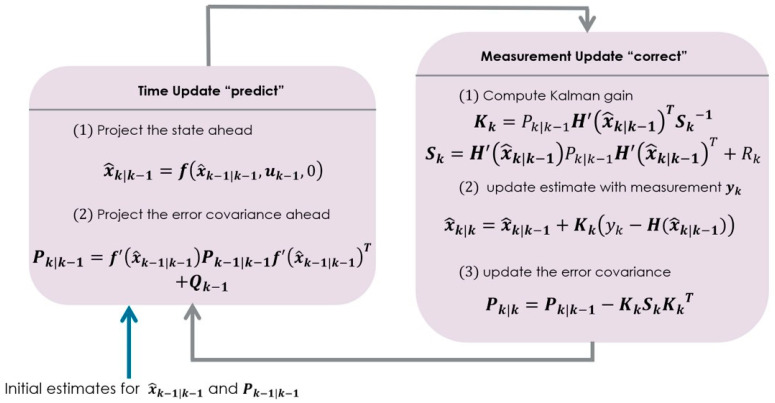
A complete picture of the operation of an extended Kalman filter.

**Figure 3 sensors-23-05877-f003:**
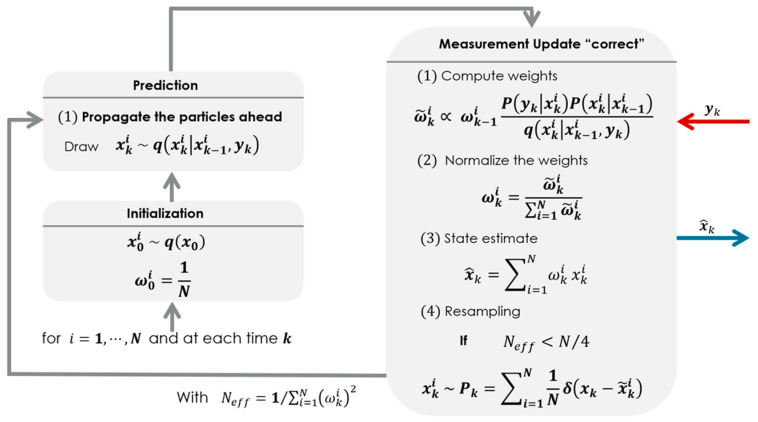
A complete picture of the operation of a particle filter.

**Figure 4 sensors-23-05877-f004:**
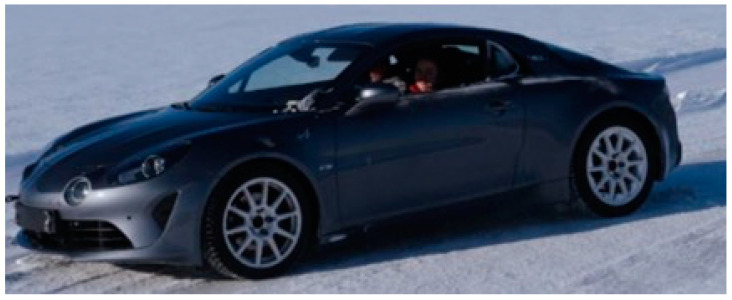
Alpine A110—Renault Sport Engineered.

**Figure 5 sensors-23-05877-f005:**
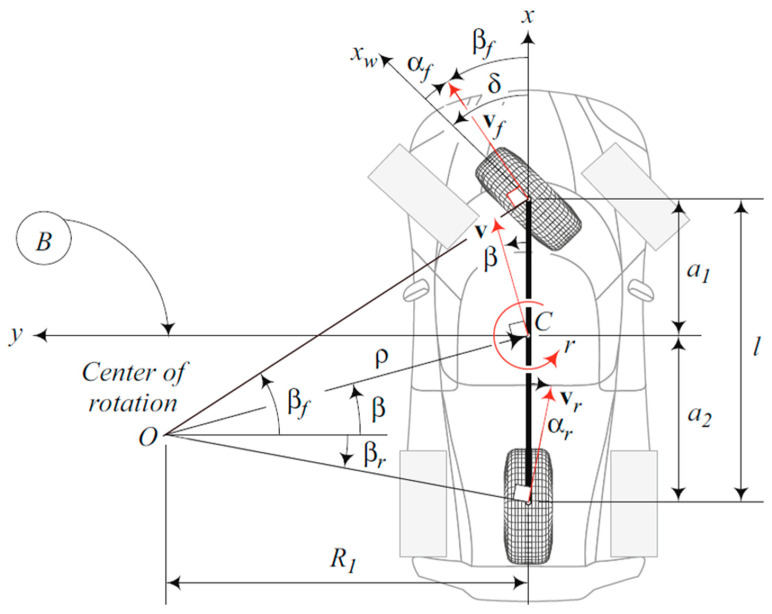
Bicycle vehicle model—Reza 2019.

**Figure 6 sensors-23-05877-f006:**
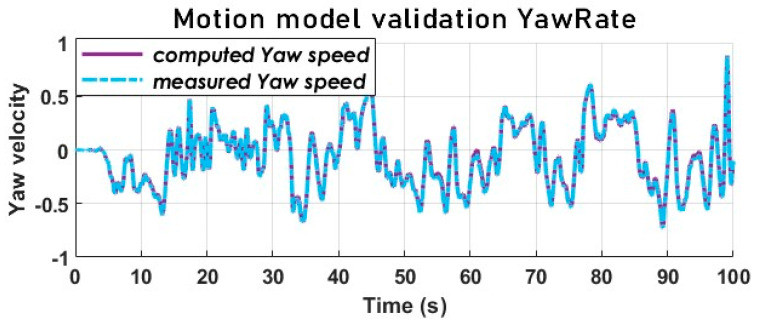
Yaw rate signals comparison-motion model validation.

**Figure 7 sensors-23-05877-f007:**
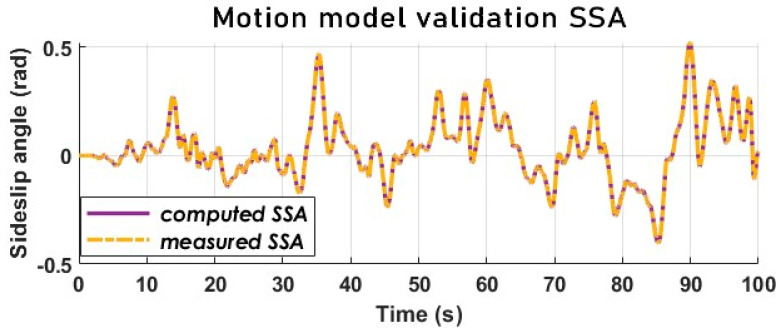
Sideslip angle comparison-motion model validation.

**Figure 8 sensors-23-05877-f008:**
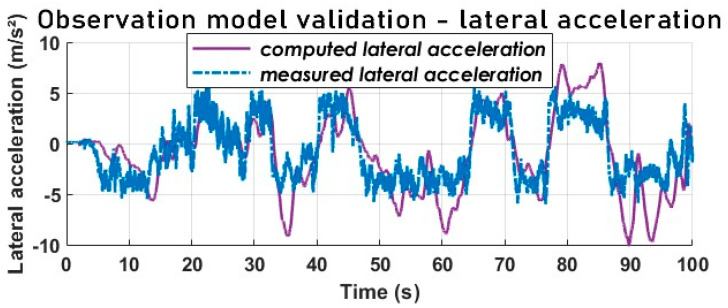
Lateral acceleration comparison-observer model validation.

**Figure 9 sensors-23-05877-f009:**
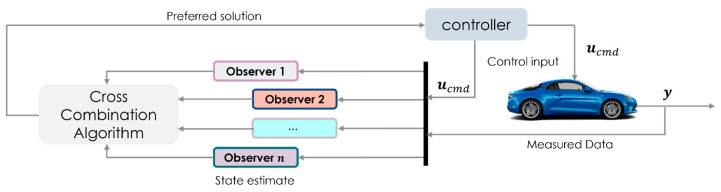
General cross-combination algorithm signals flow scheme.

**Figure 10 sensors-23-05877-f010:**
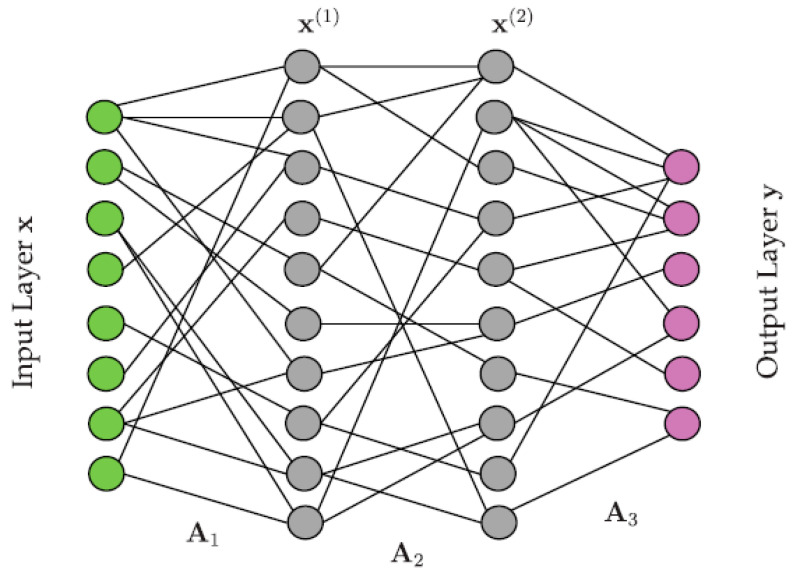
Illustration of a neural net architecture mapping an input layer x to an output layer y.

**Figure 11 sensors-23-05877-f011:**
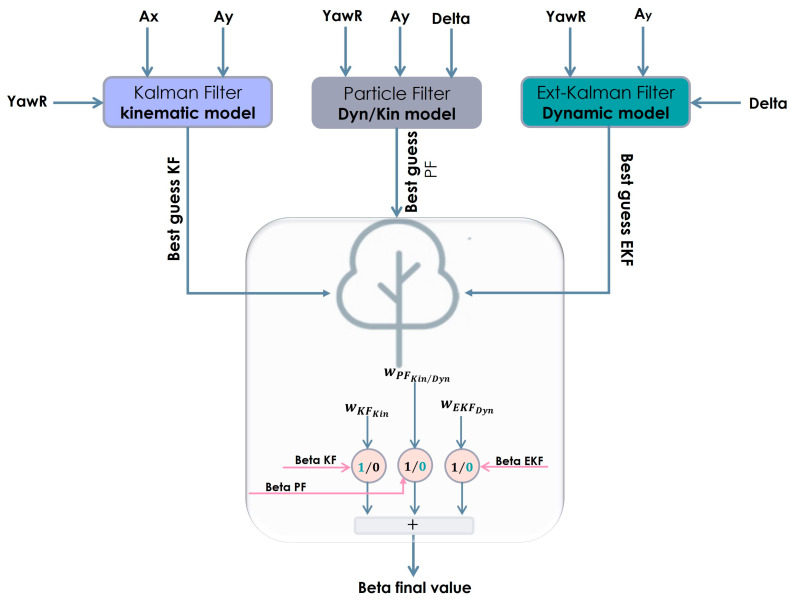
Cross-combination signals flow scheme.

**Figure 12 sensors-23-05877-f012:**
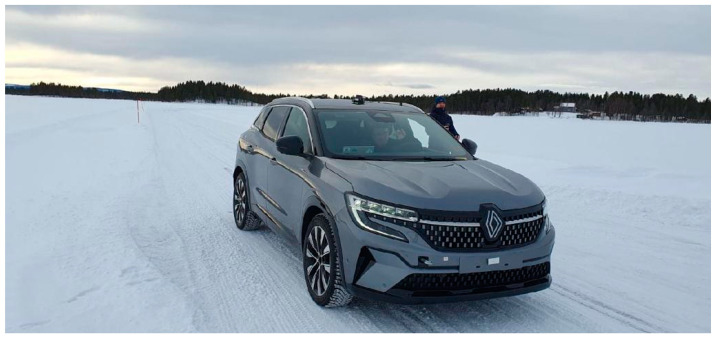
The new Renault Austral—by Renault group.

**Figure 13 sensors-23-05877-f013:**
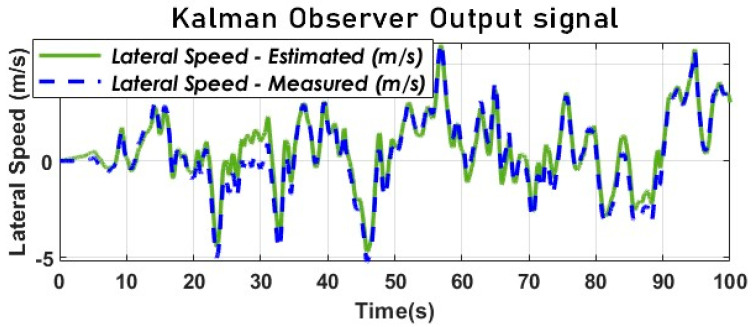
Lateral speed estimation with KF using a kinematic model (Sweden 3).

**Figure 14 sensors-23-05877-f014:**
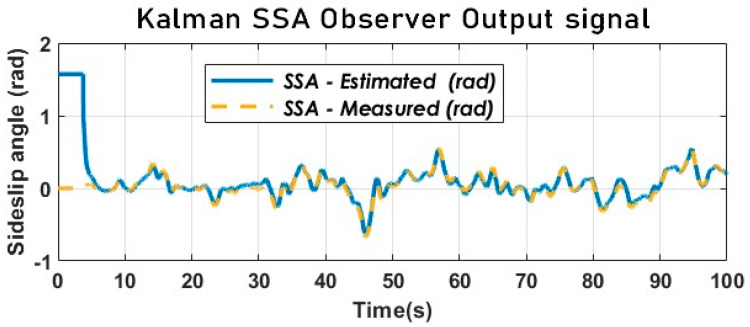
The vehicle SSA estimation with KF using a kinematic model (Sweden 3).

**Figure 15 sensors-23-05877-f015:**
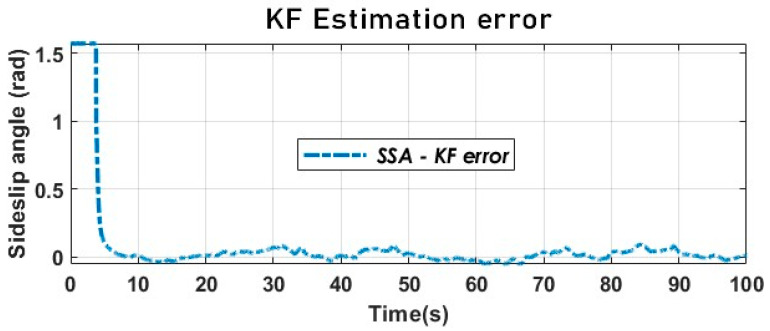
The vehicle SSA estimation error with KF using a kinematic model (Sweden 3).

**Figure 16 sensors-23-05877-f016:**
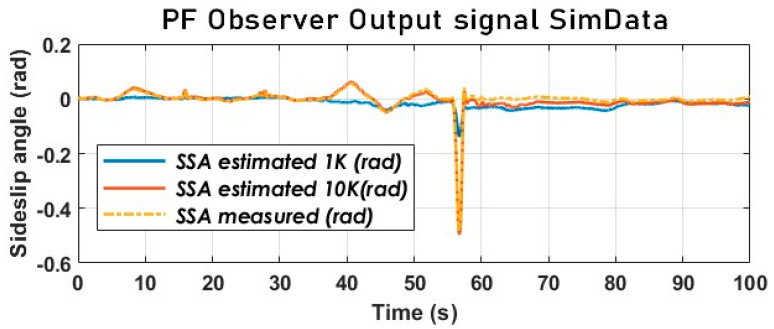
The vehicle SSA estimation with PF using a dynamic model (SimData).

**Figure 17 sensors-23-05877-f017:**
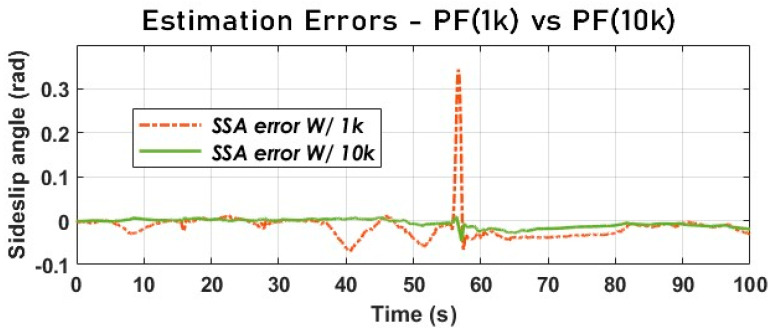
The vehicle SSA estimation error with PF using a dynamic model (SimData).

**Figure 18 sensors-23-05877-f018:**
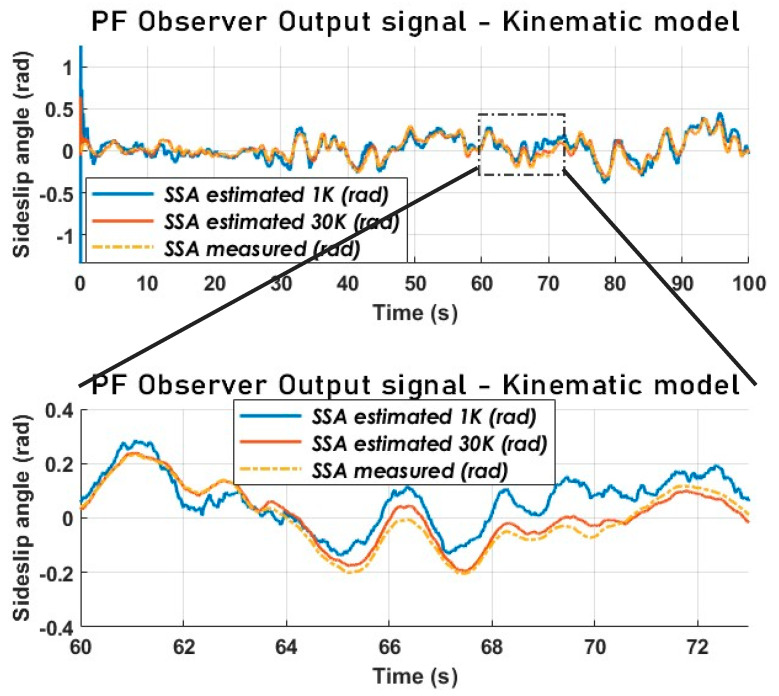
The vehicle SSA estimation with PF using a kinematic model-1 K vs. 30 K-(Sweden 4).

**Figure 19 sensors-23-05877-f019:**
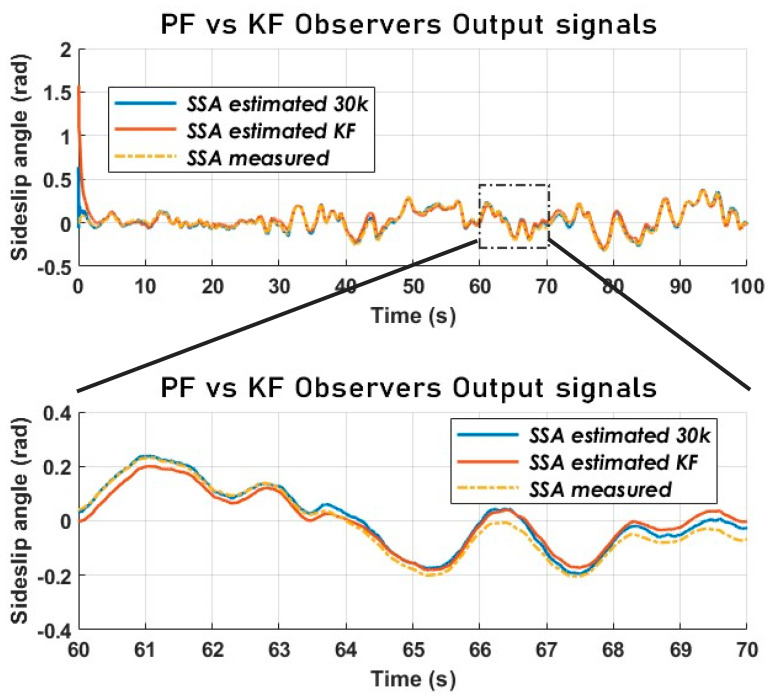
The vehicle SSA estimation KF vs. PF using a kinematic model (Sweden 4).

**Figure 20 sensors-23-05877-f020:**
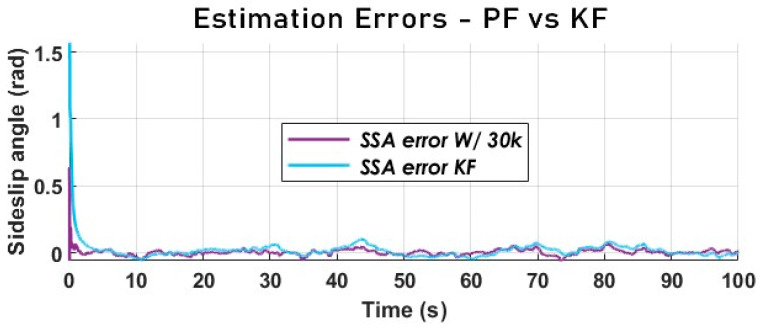
The vehicle SSA estimation error KF vs. PF using a kinematic model (Sweden 4).

**Figure 21 sensors-23-05877-f021:**
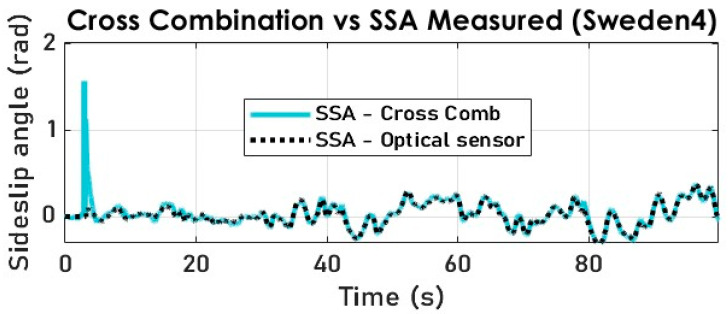
The vehicle SSA estimation with cross-combination (Sweden 4).

**Figure 22 sensors-23-05877-f022:**
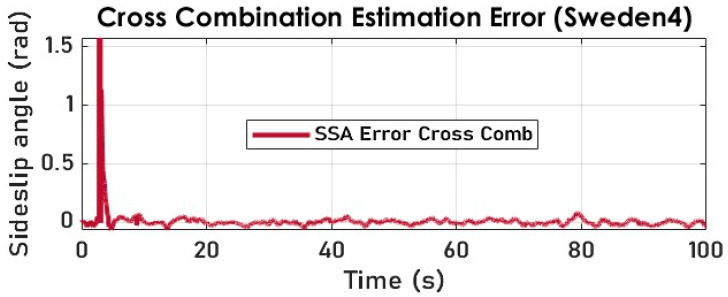
The vehicle SSA estimation error with cross-combination (Sweden 4).

**Figure 23 sensors-23-05877-f023:**
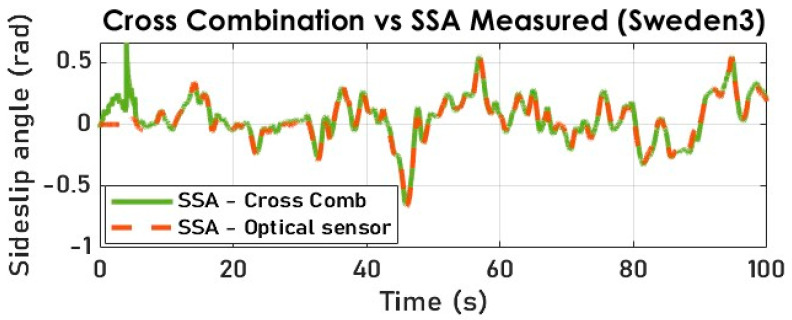
The vehicle SSA estimation with cross-combination (Sweden 3).

**Figure 24 sensors-23-05877-f024:**
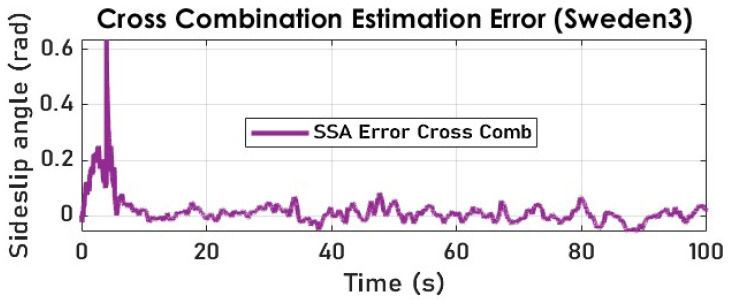
The vehicle SSA estimation error with cross-combination (Sweden 3).

**Figure 25 sensors-23-05877-f025:**
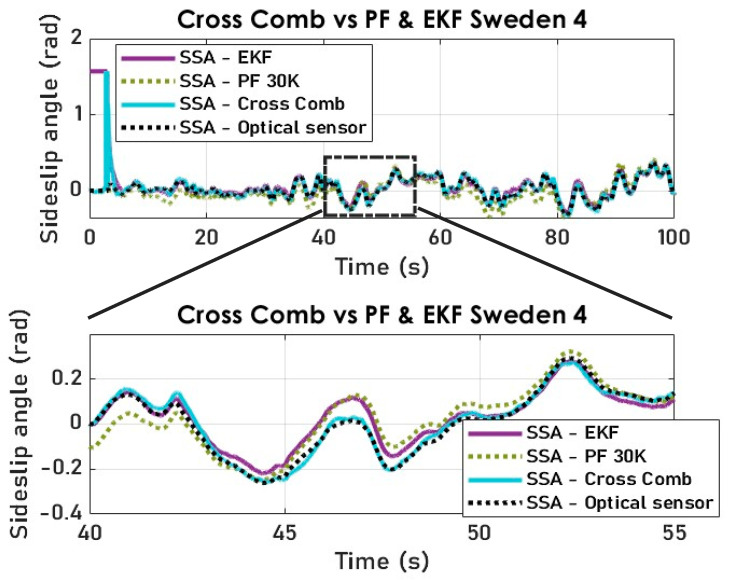
The vehicle SSA estimation comparison of PF and EKF vs. cross-combination (Sweden4).

**Table 1 sensors-23-05877-t001:** Comparison table of key performance ingredients.

Performance Index	Dataset	KF	PF	Cross-Comb
ME (deg)	Sweden 3	5.001	7.388	4.709
	Sweden 4	5.884	10.225	4.795
RMSE (deg)	Sweden 3	1.999	1.852	1.352
	Sweden 4	1.873	1.722	1.237

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
