# Peer review of "Robust Virtual Sensing of the Vehicle Sideslip Angle through the Cross-Combination of Multiple Filters Using a Decision Tree Algorithm"

_sensors, 2023, doi:10.3390/s23135877_

Round 1
Reviewer 1 Report
The authors proposed three sets of estimation approach on SSA.
1. There is no novelty in this manuscript where all the approaches implemented here were established method used for various kind of engineering processes.
2. Only one road condition were tested, should be testing in every single possible road conditions to prove robustness of the system in decision making process.
3. Very basic KF, EKF and PF. Nothing special were implemented. Theoretical framework these basic methodologies should not include in this manuscript.
4. Please check again Vx=Vx in line 262. Is that correct?
5. It would be nice if authors present block diagram of how the mathematical modelling were connected and obtained the results.
6. Include more analysis for the outcome of each output signal for different road conditions.
4. The manuscript look-like the summary of work for a thesis.
The English is understandable and appropriate terms were using.
Author Response
Dear Reviewer,
I am writing this message to express my heartfelt gratitude for your time, effort, and valuable feedback on my paper titled “Robust Virtual Sensing of Vehicle Sideslip Angle through Cross-Combination of Multiple Filters using Decision Trees Algorithm”. Your thoughtful comments and insightful remarks were extremely helpful in improving the quality and clarity of my research.
I greatly appreciate your constructive criticism and suggestions, which have helped me to refine and strengthen my work. Your expertise and attention to detail have undoubtedly contributed to enhancing the overall impact and significance of the paper.
I would like to assure you that I have carefully considered each of your comments and suggestions and have made significant revisions accordingly. Your advice has not only improved the quality of my paper but has also helped me to progress as a researcher.
Once again, I extend my sincerest gratitude to you for your dedication, expertise, and commitment to fostering scientific excellence. Your contribution as reviewer is essential in advancing knowledge and promoting rigorous scientific discourse.
Thank you for your invaluable support and for taking the time to review my paper. I am truly grateful for your commitment to the scientific community.
With warm regards,

Reviewer 2 Report
A document for suggestions about the manuscript is attached. Please follow it.

It must be improved.
Author Response

(The authors gave the same response as above.)

Reviewer 3 Report
no special comments
no special comments
Author Response

(The authors gave the same response as above.)

Reviewer 4 Report
The present paper proposes a state-of-the-art estimation technique for virtual sensing of vehicle sideslip angles through the cross-combination of multiple filters. The subject investigated is very fascinating and interesting and has a good contribution to the state-of-the-art. The methods adopted are also technically sound and additionally, the performance of the proposed system is tested using a prototype. I recommend it for acceptance following some minor revisions.
1. There are several lengthy and long statements which makes them confusing. For example, the first sentence of the abstract. It is suggested to split them into shor more meaningful sentences.
2. It is suggested to include a table of acronyms before the references as many acronyms have been used throughout the text.
3. Pros and cons of the methods proposed should be included.
4. The related works section should be further enriched by including more recent and latest references related to the subject investigated.
5. The specific contribution and novelty points should be inserted in the form of bullets for better understanding and clarity.
6. All the symbols in the formulae should be properly defined. Again a table would be useful in this regard.
7. A section describing the policy implications and usefulness of the study to field practitioners should be added.
8. Study limitations and directions for further work should be added.
9. Overall, the written language style of the paper is good, however, a thorough language audit will be useful to fix the occasional grammar issues and typos.
The use of English language is good.
Author Response

(The authors gave the same response as above.)

Reviewer 5 Report
The presented manuscript deals with an interesting topic related to a “Robust Virtual Sensing of Vehicle Sideslip Angle through two Cross Combination of Multiple Filters using Decision Trees three Algorithm”. The authors proposed a machine learning algorithm based on decision trees, connects several filters together to switch between them according to the driving context combination of three different filter using both the dynamic and kinematic vehicle model. This study certainly is interesting and contains a lot of new research and information based on conducted research.
Nevertheless, I have a few comments on the article:
By combining the dynamic and kinematic models, the proposed algorithm aims to enhance the accuracy and robustness of estimating the sideslip angle. These filters can be based on various techniques, such as Kalman filters, complementary filters, or other suitable algorithms. Multiple filters are connected together, each employing a different approach to estimate the sideslip angle. The decision tree algorithm is used to determine which filter to use based on the current driving context. The decision tree considers inputs such as vehicle speed, acceleration, steering angle, or other relevant parameters. Depending on the driving conditions and the availability of sensor data, the algorithm selects the most appropriate filter to estimate the sideslip angle accurately. Equations 18, 19, and so on,… to derive equations based on a kinematic model, you would typically start by defining the variables and parameters relevant to the system, such as the position (x, y), orientation (θ), and velocity (v) of the vehicle. These variables can be represented as functions of time (t). Figure 5 in your paper, line 290, likely provides a visual representation of the kinematic model, showing the relevant components and their relationships.
Are there limitations to the use of your method? If so, it would be worth mentioning in the conclusions?
N/A
Author Response

(The authors gave the same response as above.)

Round 2
Reviewer 2 Report
All requirements are executed generally. Also, the final status of the manuscript seems to be fine.
Reviewer 3 Report
all seems fine